# Non-Thermal Plasma Reduces HSV-1 Infection of and Replication in HaCaT Keratinocytes In Vitro

**DOI:** 10.3390/ijms25073839

**Published:** 2024-03-29

**Authors:** Julia Sutter, Jascha Brettschneider, Brian Wigdahl, Peter J. Bruggeman, Fred C. Krebs, Vandana Miller

**Affiliations:** 1Center for Molecular Virology and Gene Therapy, Institute for Molecular Medicine and Infectious Disease, Drexel University College of Medicine, Philadelphia, PA 19102, USA; js4932@drexel.edu (J.S.); jmb598@drexel.edu (J.B.); bw45@drexel.edu (B.W.); fck23@drexel.edu (F.C.K.); 2Department of Microbiology and Immunology, Drexel University College of Medicine, Philadelphia, PA 19102, USA; 3Sidney Kimmel Cancer Center, Thomas Jefferson University, Philadelphia, PA 19107, USA; 4Department of Mechanical Engineering, University of Minnesota, Minneapolis, MN 55455, USA; pbruggem@umn.edu

**Keywords:** non-thermal plasma (NTP), antiviral, herpes labialis (cold sore), HSV-1, dielectric barrier discharge, reactive oxygen and nitrogen species (RONS)

## Abstract

Herpes simplex virus type 1 (HSV-1) is a lifelong pathogen characterized by asymptomatic latent infection in the trigeminal ganglia (TG), with periodic outbreaks of cold sores caused by virus reactivation in the TG and subsequent replication in the oral mucosa. While antiviral therapies can provide relief from cold sores, they are unable to eliminate HSV-1. We provide experimental results that highlight non-thermal plasma (NTP) as a new alternative therapy for HSV-1 infection that would resolve cold sores faster and reduce the establishment of latent infection in the TG. Additionally, this study is the first to explore the use of NTP as a therapy that can both treat and prevent human viral infections. The antiviral effect of NTP was investigated using an in vitro model of HSV-1 epithelial infection that involved the application of NTP from two separate devices to cell-free HSV-1, HSV-1-infected cells, and uninfected cells. It was found that NTP reduced the infectivity of cell-free HSV-1, reduced viral replication in HSV-1-infected cells, and diminished the susceptibility of uninfected cells to HSV-1 infection. This triad of antiviral mechanisms of action suggests the potential of NTP as a therapeutic agent effective against HSV-1 infection.

## 1. Introduction

Herpes simplex virus type 1 (HSV-1) is the causative agent of herpes labialis, an oral infection characterized by the appearance of cold sore lesions. Cold sores are consequences of lytic infection when HSV-1 is actively replicating and producing infectious viruses in mucosal epithelial cells [1,2,3,4,5]. HSV-1 produced by mucosal epithelial cells can also infect sensory neurons in the trigeminal ganglia (TG), resulting in a latent infection. Latency is characterized by a substantial restriction in the number and abundance of viral transcripts as viral replication and virus production are shut down, resulting in an asymptomatic infection in patients. HSV-1 can persist in this latent state for extended periods of time until stress stimuli reactivate lytic infection, causing the re-appearance of clinical symptoms [6,7,8]. Currently, ~70% of the world’s population under the age of 50 harbors an infection with HSV-1 [9]. While the infection is typically mild, its access to the nervous system can allow for it to spread to nearby organs, causing encephalitis in the brain [10], keratitis in the eye [11], and possibly progressive neurodegenerative disorders like Alzheimer’s Disease [12,13,14,15,16,17,18,19]. The risk of developing serious neurodegenerative diseases associated with HSV-1 infection is significant in both immunocompetent and immunocompromised individuals [10,13,20].

Nucleoside analogs like acyclovir inhibit viral replication in mucosal epithelial cells and are the standard-of-care drugs for the treatment of acute HSV-1 infection, helping to alleviate clinical symptoms [21]. However, these antiviral therapies are ineffective in preventing the establishment of viral latency in the TG or clearing existing latent reservoirs in sensory neurons. Therefore, HSV-1 continues to persist in the host for the duration of their life with periodic outbreaks of cold sores [22,23]. Due to these limitations and the ongoing emergence of drug-resistant strains of HSV-1 [24], there is a critical need for new therapies that target both the lytic and latent phases of HSV-1 infection in patients. Based on the results of our investigations presented here, we propose that non-thermal plasma (NTP) can address this need.

Plasma medicine is a rapidly developing field that encompasses the biomedical use of NTP for treatment of diseases. NTP is partially ionized gas producing (vacuum) ultraviolet radiation, heat, electric fields, and chemically reactive components, including many reactive oxygen and nitrogen species (RONS) [25]. Pertinent to this study, the antiviral activity of NTP has been demonstrated against many viruses, with RONS proposed as the dominant effectors [26,27,28]. RONS are reactive and can modify structural macromolecular components of the viral particle, as demonstrated in a study where NTP modified the protein capsid of cell-free feline calicivirus (FCV) [29] and reduced virus infectivity [30]. There is also evidence of NTP-generated RONS directly modifying viral components in other human viruses, including hepatitis B virus (HBV) [31], human immunodeficiency virus type 1 (HIV-1) [32], and SARS-CoV-2 [33,34,35]. The modification of virus components may contribute to the reduced infectivity of these viruses.

Thus far, most studies of NTP against cell-free viruses have been focused on the use of NTP as a disinfection agent. While important, most of these investigations have neglected to examine the therapeutic potential of NTP against human infections by viruses like HSV-1. A study in 2014 used explanted corneal cells infected with HSV-1 as an in vitro model for herpes keratitis and showed that the indirect application of NTP resulted in an 80% reduction in HSV-1-infected cells with very little cell cytotoxicity [36]. The antiviral effect of NTP against HSV-1 was suggested to correlate with an increase in 8-oxodeoxyguanosine (8-OHdg), a marker of oxidative damage to DNA in infected corneal cells [37]. Neither study tested the therapeutic and clinical potential of NTP as a treatment for cold sores in patients living with HSV-1. We believe this paper is the first report of investigations into the use of NTP as a treatment for herpes labialis rather than a method focused solely on inactivating HSV-1.

To examine the contributions of RONS as antiviral effectors, we used two devices: a two-dimensional (2D)-DBD device, which delivers large concentrations of RONS to targets and has a demonstrated antiviral effect against different viruses [30], and a floating electrode dielectric barrier discharge (FE-DBD) device, which delivers its biological effects through effectors that include short- and long-lived RONS and electromagnetic fields associated with plasma generation [38]. In cells, RONS are naturally generated by organelles during metabolism, signaling, and immune responses and neutralized by antioxidants to maintain cellular redox homeostasis [39,40,41,42,43]. During HSV-1 infection, the virus destabilizes the redox homeostasis by hijacking cellular oxidative stress pathways to craft an environment favorable for their replication and to mitigate immune clearance [44,45,46]. NTP-generated RONS can shift the redox environment back in favor of the cell, counteracting viral evasion strategies and potentially promoting HSV-1 clearance [45]. Outside the cell, NTP-generated RONS can oxidize proteins and lipids on the surfaces of viruses and cells.

Our studies demonstrated the antiviral effect of direct NTP application on cell-free HSV-1 using two different devices, which were characterized based on their production of long-lived RONS. In addition, NTP reduced viral replication in HSV-1-infected cells and the susceptibility of uninfected cells to HSV-1 infection.

## 2. Results

A herpes labialis lesion contains three targets of a potential NTP-based therapy: cell-free HSV-1 virions (viruses released by infected cells into the extracellular spaces), HSV-1-infected epithelial cells (cells hosting active and productive HSV-1 replication), and uninfected epithelial cells (i.e., cells not yet infected but susceptible to infection by virus produced by nearby infected cells). We hypothesized that the clinical benefits of an NTP-based therapy will ideally be realized from effects on all three targets. First, NTP will damage cell-free HSV-1 virions and reduce their infectivity (i.e., the ability to infect susceptible cells). Second, NTP treatment will interfere with active replication in infected cells, thereby reducing the quantity of viruses released into the extracellular spaces. Third, the application of NTP to uninfected cells will reduce their susceptibility to infection by extracellular viruses and therefore curtail the spread of infection in the lesion. Our in vitro model of HSV-1 epithelial infection and NTP application was used to test each aspect of our hypothesis using the direct application of NTP to cell-free HSV-1, HSV-1-infected cells, and uninfected cells. In this model, the KOS-GFP-HSV-1 strain, which includes a green fluorescent protein (GFP) gene in place of a viral immediate early (IE) gene, was used to quantify HSV-1-infected cells through GFP expression as a reporter of infection [47]. This recombinant HSV-1 strain is fully infectious and shows no appreciable growth defects during in vitro infection. In addition, the infection of mice with this strain is characterized by virus spread, the establishment of latent neuronal infection, reactivation, and virus-associated pathogenesis.

### 2.1. NTP Exposure Reduces the Infectivity of Cell-Free HSV-1

NTP devices can be used to produce antiviral effects against many viruses [26,30]. To investigate the effects of FE-DBD and 2D-DBD on HSV-1 infectivity, we established operating conditions to deliver NTP to cell-free virus suspensions (Figure 1A). The FE-DBD plasma reduced the infectivity of cell-free HSV-1 in a frequency-dependent manner to 89% at 400 Hz and a statistically significant 70% at the highest frequency (1000 Hz) (Figure 1B). Mean fluorescent intensity (MFI), which is an indicator of viral gene expression levels, decreased with increasing NTP frequency from 86% at 400 Hz to 66% at 1000 Hz (relative to the infected control) (Figure 1C). A similar dose-dependent trend toward reduced viral infectivity was observed after exposure to 2D-DBD plasma (89% infectivity after a 4 min exposure and a reduction to 63% at 7 min) (Figure 1D). Similarly, MFI trended down with increased exposure time (80% at 4 min to 66% at 7 min, relative to the infected cell control) (Figure 1E). These results indicate that the direct application of FE-DBD or 2D-DBD to a cell-free virus adversely impacts the subsequent infection of target cells.

### 2.2. NTP Disrupts Viral Replication and Gene Expression in HSV-1-Infected HaCaT Cells

While the antiviral effect of NTP on cell-free viruses has been shown [26,30], few studies have investigated the impact of NTP on productive viral infection in cells [36,48]. To examine this effect, HSV-1-infected HaCaT cells were subjected to NTP application after infection (Figure 2A). Three MOIs were used for the infection of HaCaT cells to determine the effectiveness of NTP against virus replication.

We elected to expose infected cells to FE-DBD at a frequency of 400 Hz because the viability of uninfected cells was unaffected by NTP exposure at this frequency (Figure 2B). At 0.5 MOI, the number of HSV-1-infected cells was significantly reduced by post-infection NTP application (~70% reduction relative to cells infected without subsequent NTP application). At 0.1 MOI, post-infection NTP application appeared to reduce the number of HSV-1-infected cells to an even greater extent (>70% reduction relative to control cells). However, this reduction was not statistically significant, likely due to a low number of infected cells at that MOI. When cells were infected at the highest virus inoculum (MOI 1), no significant changes in the number of infected cells were measured 24 h post-NTP application (Figure 2C). As an additional indicator for NTP antiviral activity, MFI followed similar trends with a significant NTP-associated reduction measured at 0.5 MOI (Figure 2D). Our results suggest that NTP application can adversely affect HSV-1 replication post-entry. Since GFP expression is a measure of IE promoter activity, reductions in MFI suggest that the inhibition of IE viral gene expression may be part of the mechanism of action of NTP.

A 7 min NTP exposure was selected for the application of 2D-DBD plasma to infected cells, since this exposure duration yielded no significant changes in the viability of uninfected cells (Figure 2E). While similar trends of fewer infected cells (Figure 2F) and lower MFI (Figure 2G) in response to NTP were observed at each MOI, no statistically significant changes were measured.

### 2.3. NTP Exposure Impacts the Susceptibility of HaCaT Cells to Infection with HSV-1

Newly released HSV-1 virions spread from a productively infected cell to neighboring uninfected cells by diffusion through the extracellular space or by spreading cell to cell via cellular junctions [49]. As targets in cold sores, uninfected epithelial cells that are rendered less susceptible to HSV-1 by NTP treatment could be an important aspect of reducing the local spread of HSV-1 within the lesion, hastening virus clearance, and lesion resolution. To examine the ability of NTP to reduce the susceptibility of cells to infection, uninfected cells were exposed to NTP at the selected doses and then infected 1 h later with HSV-1 at three MOIs.

When uninfected HaCaT cells were exposed to FE-DBD, the number of infected cells was reduced by >70% during subsequent infections at 0.1 and 0.5 MOI relative to their respective controls. The reduction at 0.5 MOI was statistically significant, while the observed reduction at 0.1 MOI was not. Additionally, the number of infected cells was significantly reduced (26% reduction) at 1.0 MOI (Figure 3B). These reductions suggest that NTP exposure inhibits the infection of keratinocytes by HSV-1. MFI was decreased by ~70% at 0.5 MOI and 22% at 1 MOI, with a reduction trend at an MOI of 0.1 (Figure 3C). This result suggests that NTP-exposed cells, after becoming infected, have reduced levels of IE viral gene expression and altered susceptibility to HSV-1 binding and/or entry. This result is consistent with observations made in experiments involving NTP application to cell-free HSV-1 (Figure 1). While the application of 2D-DBD resulted in similar trends toward reduced infection (Figure 3D) and apparently lower viral gene expression (Figure 3E), the effects of 2D-DBD were not statistically significant under the conditions of the experiments.

### 2.4. NTP Produces Long-Lived RONS in Media during Exposure

To establish possible relationships between the observed antiviral effects of NTP and the RONS generated by NTP, two long-lived RONS—hydrogen peroxide and nitrite—were measured in cell culture medium with and without HaCaT cells subsequent to NTP exposure. Hydrogen peroxide and nitrite concentrations were measured immediately after NTP exposure and 24 h post-NTP exposure to establish correlations with the antiviral effects observed at 24 h. Target cells (HaCaT keratinocytes) were introduced to investigate the contributions of cellular redox systems, which include enzymes and other molecules that produce or inactivate RONS to maintain cellular redox homeostasis [38,45], to changes in RONS concentrations.

In experiments involving FE-DBD plasma (Figure 4A–D), there was an overall increase in hydrogen peroxide immediately after NTP application at all frequencies tested relative to the no-NTP controls, ranging from 38 μM to 50 μM (Figure 4A). These concentrations did not change significantly over 24 h (Figure 4B). When HaCaT cells were introduced into the system, no apparent change in the concentration of hydrogen peroxide was observed at the 0 h timepoint (Figure 4A). However, at 24 h post-NTP exposure, the hydrogen peroxide concentrations exceeded 200 μM in media containing HaCaT cells in all cases, even in cells not exposed to NTP. This suggests that the cells themselves produce considerable quantities of hydrogen peroxide over time independent of NTP (Figure 4B).

In contrast to FE-DBD, the concentrations of hydrogen peroxide in media alone were higher when 2D-DBD was applied, ranging from 140 μM to 600 μM in an exposure duration-dependent manner and reduced by over 50% 24 h later. In the presence of HaCaT cells, the hydrogen peroxide concentrations were observably lower than in media alone at 400 and 1000 Hz (>40% reduction) (Figure 4E). While the no-NTP control cells had 30 μM hydrogen peroxide after 24 h incubation, the hydrogen peroxide concentrations of the samples exposed to NTP ranged from 240 μM to 270 μM. However, no exposure duration-dependent trend was observed (Figure 4F).

The nitrite concentrations immediately after FE-DBD NTP application were frequency-dependent and ranged from 9 μM to 17 μM. When HaCaT cells were present, the nitrite concentrations decreased by a statistically significant value of ~40% for all groups (Figure 4C). After 24 h, the nitrite concentrations ranged from 7 μM to 20 μM, with there being no statistical significance between groups (Figure 4D). While nitrite and reactive nitrogen intermediates have documented antimicrobial activities, this effect is pH-dependent [50]. Our previous studies have shown that FE-DBD NTP application does not change the pH of media [38].

2D-DBD plasma also produced higher concentrations of nitrite (>50 μM) compared to FE-DBD at both time points. There was no dependence of the nitrite concentrations on exposure duration immediately after NTP exposure. The nitrite concentrations in the absence of cells ranged between 96 μM and 135 μM. When HaCaT cells were present, the nitrite concentrations appeared to increase. However, the differences were not statistically significant (Figure 4G). After 24 h, the nitrite concentrations ranged from 82 μM to 102 μM in media, with no trend in exposure duration. While the presence of HaCaT cells appeared to increase the nitrite concentrations, this increase was only statistically significant with the 6 min NTP exposure (*p* = 0.0438) (Figure 4H).

## 3. Discussion

Herpes labialis is a lifelong disease with no available curative strategies, resulting in a large global health burden [9,51]. The current antiviral therapies involve the pharmacological use of nucleoside analogs to inhibit HSV-1 infection by targeting the replication of its double-stranded DNA genome [52]. While these drugs, such as acyclovir, are modestly effective in alleviating cold sores, they fail to eliminate latent reservoirs that cause recurrent reactivation in patients [22,23]. This limitation, combined with emerging antiviral resistance [24], highlights a critical need for the development of new therapies that can suppress viral replication in cold sores while also reducing latent reservoirs in the TG and subsequent recurrences of clinical manifestations of HSV-1 reactivation.

Using our in vitro model of epithelial cell HSV-1 infection, we investigated NTP as the basis of a therapy for herpes labialis. Our experiments demonstrated that the antiviral activities of NTP include (i) reductions in virus infectivity, (ii) reductions in cells infected with HSV-1, (iii) reductions in viral gene expression during replication in host cells, and (iv) reductions in the susceptibility of uninfected cells to HSV-1 infection.

The first result was expected given previous demonstrations of the detrimental effects of NTP on cell-free viruses. As an enveloped virus, HSV-1 surrounds its capsid and enclosed double-stranded genome with an envelope composed of a lipid bilayer and membrane-embedded glycoproteins. Enveloped viruses are generally more susceptible to damage by environmental stresses (e.g., heat, desiccation, and oxidation) than non-enveloped viruses due to the relative fragility of the envelope and exposure of the envelope proteins and lipids to the outside environment. The reductions in HSV-1 infectivity demonstrated in our experiments are suggestive of NTP-associated oxidative damage to the envelope lipids and glycoproteins in the virus structure. They are also consistent with other studies of virus damage and associated reductions in virus infectivity caused by NTP exposure [5,29,30,31,32,33,53].

While reductions in infected cell numbers subsequent to the direct application of NTP to a cell-free virus are likely attributable to NTP-associated modifications in viral glycoproteins involved in attachment and entry into target cells, dose-dependent reductions in MFI suggest that other components within the virus particle were impacted. In these experiments, the MFI of GFP reporter expression was a measure of viral gene expression in individual HSV-1-infected cells. Speculatively, the modification of the viral DNA genome by NTP-delivered RONS (without damage to envelope glycoproteins) could have impaired post-entry viral gene expression, resulting in less robust virus replication (i.e., lower MFI) in cells infected by viruses with damaged genomes.

The application of FE-DBD NTP (at 1000 Hz) to a cell-free virus suspension resulted in a modest but significant reduction in infectivity (Figure 1B) rather than a multi-log reduction typical of more effective antimicrobial agents (such as heat, detergents, or other disinfecting agents). In previous studies involving the application of NTP to a cell-free virus preparation [29], a 15 s application of NTP reduced the infectivity of calicivirus (suspended in 100 µL of medium) to less than one log relative to unexposed virus controls. Our modest 40% reduction in HSV-1 infectivity (in a 4-fold larger volume of medium) after a 20 s exposure to FE-DBD plasma was comparable despite differences in virus identity, suspension medium, and virus suspension volume relative to the calicivirus experiments. We also chose to place an upper limit on NTP delivery in each of the three types of experiments so that we could examine NTP antiviral activity in cells without the complication of NTP-associated cytotoxicity. We are confident that we would have observed multi-log reductions in virus infectivity had we chosen to expose cell-free HSV-1 to FE-DBD NTP for the extended durations used in the calicivirus studies (up to three minutes).

The second result parallels and extends previous experimental results involving the application of NTP to HSV-1-infected corneal epithelial cells [36]. However, the third and fourth activities are unique among published studies of the effects of NTP on viruses and viral infections. These activities are also unique to NTP compared to other forms of photodynamic therapy (PDT), which has been investigated as a tool for inactivating virus inoculums [54]. Unlike the application of NTP, however, the pre-treatment of target cells with PDT had no impact on HSV-1 infection [55]. Furthermore, studies involving the application of PDT to infected cells remain inconclusive [54].

Our experiments involving NTP application to HSV-1-infected cells demonstrated that NTP reduced the number of infected cells as well as the levels of reporter expression within infected cells. However, our results also indicated that both effects were dependent on the magnitude of the infection (i.e., MOI). The greatest reductions in the HSV-1-infected cells and MFI in response to NTP were observed at MOIs of 0.1 and 0.5. In contrast, the antiviral activities of NTP against cells infected at an MOI of 1 were negligible. The dependence of antiviral activity on MOI suggests that the availability of the NTP-associated factor responsible for the antiviral activity may be rate-limiting, as implied by the loss of activity in the presence of a high percentage of HSV-1-infected cells. Although this result may inform an understanding of the mechanisms that underlie the antiviral activities, it does not diminish the potential of NTP as a therapy for HSV-1 infection since a condition equivalent to an MOI of 1 is unlikely in a lesion typical of herpes labialis.

The effect of NTP on GFP expression at the single cell level suggests one or more NTP associated mechanisms that affect viral gene expression. After virus entry into the cell, viral gene expression during productive replication proceeds through three sequential and cascading phases: the immediate early (IE), early (E), and late (L) phases [23,52]. The GFP reporter in the virus used in these studies signals viral gene expression in the IE phase. Our results suggest that NTP interferes with IE viral gene transcription. These viral genes are important for producing viral proteins that downregulate cellular antiviral responses as well as viral gene expression cascades that promote progression through the viral replication cycle. The disruption of IE gene expression can lead to a reduced expression of early (E) genes and late (L) genes that encode protein products for viral replication and assembly, respectively [52]. It is also possible that other mechanisms are acting to affect reporter activity. Future studies have been planned to parse out the mechanisms that underlie the effect(s) of NTP on events across the HSV-1 replication cycle.

Disrupting viral gene expression in the host cell will result in less virus production, fewer infected cells at the site of infection, and a lower virus titer in the lesion. By reducing lytic infection locally, NTP treatment can reduce the titer of HSV-1 available for infecting adjacent mucosal epithelial cells and, importantly, for establishing latent infection in sensory neurons. These effects will be verified in a murine model of HSV-1 oral infection and pathogenesis, in which quantitative polymerase chain reaction (qPCR), Western blot, and viral plaque assays will be used to verify the effects of NTP on in vivo viral gene expression, protein expression, and virus production, respectively [2]. The murine model will also allow us to test the hypothesis that a reduction in HSV-1 production minimizes the establishment of latent reservoirs in the TG and, therefore, reduces the likelihood and/or frequency of reactivation and recurrent outbreaks of cold sores over time [45].

We also observed a reduced susceptibility to HSV-1 in uninfected cells exposed to NTP. It is likely that the reduced susceptibility of HaCaT cells to HSV-1 is due to the oxidation of cell surface receptors involved in HSV-1 entry, such as heparan sulfate. This phenomenon was demonstrated in glycoengineered keratinocytes in which modified glycan residues negatively impacted HSV-1 entry and spread [56]. A similar mechanism was implied in experiments involving NTP and HIV-1 infection. In those studies, NTP altered the expression of extracellular receptors involved in viral entry during the in vitro infection of HIV-1 [32].

The proposed NTP-based treatment is not limited to the NTP-mediated disinfection or sterilization of isolated viruses. In a clinical setting in which NTP is used to treat a cold sore, multiple mechanisms of antiviral activity provided by NTP, as demonstrated in our studies, will combine to increase the efficacy of the treatment. First, reductions in the infectivity of cell-free viruses within the lesion will be achieved, presumably by the modification of glycoproteins and lipids found on the surface of HSV-1 viral particles and involved in entry into the host cell. Second, reductions in viral gene expression will be achieved in cells already infected at the time of treatment. Third, NTP will reduce the susceptibility to infection of the cells not yet infected in the lesion. Collectively, these effects will serve to reduce the viral titer in the lesion, reduce the local spread of the virus, and decrease the local pathogenesis associated with infection. This proposed therapy, with its multi-faceted antiviral activities, has a distinct advantage over currently used drugs, which act through just a single mechanism of action.

We also propose that a reduction in local virus titers will translate into a reduction in the infection of neurons that innervate the lesion and the establishment of latent infections in those neurons. We hypothesize that reductions in virus reservoirs in latently infected neurons will consequently reduce the frequency of recurrent herpes labialis caused by the reactivation of infection from latently infected neurons. We are now extending our investigations to a murine model of oral herpes to demonstrate the effect of NTP treatment on the neuronal viral reservoir.

Through the use of two different NTP devices, we demonstrated that the treatment modality determines the efficacy of NTP and may be attributed to the individual experimental set-ups required. The electrode used to generate FE-DBD was designed to fit inside a well in a 12-well plate (6 × 10^5^ cells/well) and deliver NTP at a distance of 1 mm from cells on the bottom surface of the well. In contrast, the 2D-DBD electrode rested on top of a 96-well plate where the cell density was 7 × 10^4^/well and the distance between the electrode and cells was ~14 mm. The disparity in the number of cells exposed to NTP for each device and the distance from the electrode may have contributed to some of the differences in our experimental observations.

Additional key differences between the two devices were the presence or absence of cell culture medium during NTP application to cells and the electrical characteristics of each device design. FE-DBD plasma was delivered after the removal of medium from the well to allow for a more “direct” application. During application the of 2D-DBD plasma, the medium remained on the cells during NTP exposure to prevent cell loss due to dehydration during the extended exposures. Additionally, in the case of FE-DBD plasma, the target becomes part of the electrical circuit and is therefore exposed to short-lived metastable species, charged species, and electric fields, directly capturing all of the NTP effectors [38]. In contrast, the 2D-DBD device mostly delivers long-lived RONS to the target cells, and the target cells are not part of an electrical circuit [30].

We also measured the RONS produced in the NTP generated by the two devices, as the concentrations of RONS have been shown to differ between NTP modalities [26]. The hypothesis driving our studies was that changes in hydrogen peroxide and nitrite concentrations, which differ with device, correlate with NTP-associated antiviral activities demonstrated in experiments involving cell-free viruses, infected cells, and uninfected cells [57]. Our study showed that the antiviral activity of NTP did not correlate with the concentration of RONS delivered by either device. Despite its delivery of greater concentrations of RONS to the sample, 2D-DBD application resulted in less antiviral activity. In contrast, the FE-DBD delivered lower RONS concentrations but had greater antiviral activity. These results suggest that other NTP effectors and the more direct contact with the sample with the FE-DBD likely explain the higher antiviral effectiveness against HSV-1 infectivity and infection.

RONS are proposed to be the dominant effectors in NTP’s antiviral effects [26]. 2D-DBD, previously shown to have antiviral activity against cell-free FCV [30], is characterized by the generation of long-lived RONS. A 7 min 2D-DBD NTP exposure, which was applied to HSV-1-infected and uninfected cells, produced ~596 μM hydrogen peroxide and ~102 μM nitrite in media. Under these treatment conditions, 2D-DBD did not significantly reduce HSV-1 replication in infected cells or the susceptibility of uninfected cells to HSV-1. At 400 Hz (50 μM hydrogen peroxide and 28 μM nitrite), FE-DBD NTP exhibited greater efficacy against HSV-1-infected cells and uninfected cells. Together, these observations suggest that long-lived RONS are not primarily responsible for NTP’s antiviral mechanism against HSV-1 infection. Some studies have proposed a biological contribution of other NTP components, including short-lived RONS, charged species, or electric fields [25,58]. In the 2D-DBD, these components would not reach the cells when the 2D-DBD NTP is delivered, weakening the efficacy of NTP. Under our experimental conditions, the antiviral effectiveness was device dependent but did not correlate with the concentration of RONS measured.

Overall, these initial studies provide the first insights into a potential NTP-based therapy against oral HSV-1 infection. We showed that NTP reduced cell-free HSV-1 infectivity, diminished viral gene expression, and reduced the susceptibility of uninfected cells to HSV-1. Based on these results, we conclude that the combined antiviral effects of NTP could lead to reductions in the spread of HSV-1 to nearby cells in an infected lesion, as well as to the TG, where the virus establishes latent infection. This would reduce the frequency of viral reactivation and, therefore, lead to reductions in cold sore outbreaks.

Future studies will expand on these initial efforts. Our investigations highlighted three distinct antiviral effects of NTP application: reductions in cell-free virus infectivity, the disruption of virus replication and viral gene expression in infected cells, and the diminished susceptibility of host cells to infection. Each effect likely has a different underlying mechanistic explanation: diminished virus integrity, stress responses in infected cells, and altered cell surface molecules that support virus binding and entry, respectively. Upcoming mechanism of action investigations will connect the antiviral effects to their underlying molecular mechanisms of action.

Additionally, the safety and effectiveness of an NTP-based therapy for HSV-1 infection will be demonstrated using a mouse model that can replicate cold sore formation using the lip scarification method of infection [2]. This model will allow us to examine the resolution of cold sores and the impact of NTP on lesion resolution, virus titer in the lesion, and the establishment of latency in innervating nerves. In addition, the mouse model of NTP treatment for oral HSV-1 infection will facilitate investigations of the immunomodulatory impact of NTP on HSV-1 infection, spread, and clearance. NTP can stimulate the local release of immunogenic molecules from exposed cells as well as innate immune responses in exposed cells. We have previously demonstrated this effect in a lymphocyte cell line that models latent HIV-1 infection [48] and in mice bearing colorectal tumors [59]. The immunomodulatory activity of NTP further distinguishes NTP from current anti-HSV-1 pharmaceutical treatments which do not act through the host immune system. The role of NTP-induced innate and adaptive immune responses in clearing acute HSV-1 infection and abating recurrent infections will be explored in this mouse model.

As seen in our in vitro model, the efficacy of NTP is dependent on the optimal dose and the device selected for treatment. While our initial studies aimed to determine the optimal dose by measuring the RONS delivered by NTP, we found that the RONS concentrations did not correlate with antiviral activity. Instead, an optimal NTP dose would coincide with the desired biological effectors observed in cells or in vivo. Therefore, an optimal dose should be capable of disrupting HSV-1 replication at the lesion site, preventing the establishment of latent infection, and promoting antiviral responses against HSV-1 antigens for immune control over infection. In addition, investigations into which effectors are responsible for this antiviral activity need to be carried out to further aid in the selection of an optimal dose for treatment.

Our initial investigations of the effect of NTP on early events in HSV-1 infection, as well as other studies of NTP as an immunomodulatory agent, suggest that NTP could serve as a therapeutic alternative that can address both lytic and latent infection and will stimulate a specific anti-HSV-1 immune response that will extend the effectiveness of NTP treatment well beyond the short duration of the treatment.

## 4. Materials and Methods

### 4.1. HaCaT Cell Culture

HaCaT adult human keratinocytes (AddexBio Technologies, San Diego, CA, USA) [60] were cultured in Dulbecco’s Modified Eagle Medium supplemented with 10% fetal bovine serum (FBS), 2 mM l-glutamine, and 1% penicillin/streptomycin (DMEM10). Cells were maintained at 37 °C, 5% carbon dioxide (CO_2_), and 95% relative humidity and passaged every 3–4 days.

### 4.2. HSV-1 Infection

The KOS-GFP-HSV-1 strain (Figure 5A), which contains a GFP marker in place of the ICP4 gene under the control of a cytomegalovirus (CMV) promoter [47], was prepared at a stock concentration of 3 × 10^7^ PFU/mL. Viral inocula were prepared with serum-free DMEM at multiplicities of infection (MOI; number of viral particles per cell) of 0.1, 0.5, and 1.0 relative to the number of cells plated in each well. HaCaT cells were infected with 100 μL of the viral inoculum for 1 h at 37 °C and 5% CO_2_ with rocking. After the 1 h infection, the inoculum was removed and the cells were supplemented with new DMEM10.

### 4.3. NTP Devices

An FE-DBD 12-well electrode connected to an electrical power supply was used to investigate the antiviral activity of NTP on HSV-1 infection. This 12-well electrode consists of a copper electrode surrounded by quartz dielectric material casing. A Z-positioner was used to set the distance of the electrode to 1 mm from the bottom of the 12-well plate placed on a metal ground plate (Figure 6A). The voltage, frequency, and exposure time were set on the power supply.

For comparison, a 2D-DBD device (described in detail in [30]) was also used. This electrode consists of a 2D array of micro-discharges encased in a polytetrafluoroethylene electrode holder that facilitates the continuous flow of ambient air at a rate of 15 slm (standard liter per minute). The electrode was driven by an AC power supply at a frequency of 30 kHz. The applied voltage amplitude was monitored with a high voltage probe through an oscilloscope and kept at 5 kV, corresponding to an average power of ~5 W. The distance between the 2D-DBD array and the sample was controlled using an adjustable platform (Figure 6B). The electrode was centered over a 3 × 3 grid of wells in a 96-well plate, resulting in the simultaneous exposure of all nine wells in the grid to the NTP airflow.

### 4.4. Application of NTP

For NTP application using the FE-DBD, HaCaT cells were seeded at 6 × 10^5^ cells/mL in 1 mL of DMEM10 into each well of a 12-well plate. Cells were grown to confluence over a 24 to 26 h period. The cells were washed with PBS, and then the 12-well plate was fitted with a metal grounding plate. Following the removal of the PBS from the cells, the electrode was placed 1 mm above the bottom of each well using an electrode holder with an XY positioner. NTP was applied to the HaCaT cells for 20 s at 8.5 kV and 400 Hz frequency. Immediately after NTP exposure, the cells were supplemented with 1 mL of DMEM and incubated at 37 °C and 5% CO_2_ for 24 h. For the HSV-1-infected cells, NTP exposure took place immediately after the 1 h HSV-1 infection period. Uninfected HaCaT cells were exposed to NTP and infected with HSV-1 after 1 h of incubation.

For the exposure of cell-free HSV-1 to FE-DBD, a viral inoculum at a concentration of 6 × 10^6^ PFU/mL was prepared in 1 mL of serum-free DMEM. NTP was applied to 400 μL of this inoculum in a well of the 12-well plate at a distance of 2 mm from the bottom of the well for 20 s at 8.5 kV and frequencies of 400 Hz, 500 Hz, and 1000 Hz. Immediately after NTP exposure, 100 μL of the viral inoculum (MOI 1) was added to confluent HaCaT cells in a 12-well plate (plated at 6 × 10^5^ cells per well). After allowing for infection to proceed for 1 h, the inoculum was removed, and cells were supplemented with 1 mL new DMEM10 and incubated at 37 °C and 5% CO_2_ for 24 h.

For the 2D-DBD device, HaCaT cells were seeded in 96-well plates at a seeding density of 7 × 10^4^ cells/well in 100 μL DMEM10 and grown to confluence over a 24 to 26 h period. Prior to NTP exposure, the cells were washed with PBS, and 100 μL of fresh DMEM10 was added to each well. The base of the electrode housing was lowered to touch the top of the 96-well plate. This resulted in a distance of approximately 14 mm between the electrode array and the bottom of the well. The HaCaT cells in DMEM10 were exposed to NTP for 7 min. For the HSV-1-infected cells, NTP exposure took place immediately after the 1 h HSV-1 infection period. One-hour post-NTP exposure, the media were replaced with new DMEM10 (not exposed to NTP). Uninfected HaCaT cells were exposed to NTP and infected with HSV-1 after 1 h of incubation.

In experiments involving the application of 2D-DBD plasma to cell-free HSV-1, the virus was suspended in serum-free DMEM at a concentration of 7 × 10^5^ PFU/mL and exposed to NTP for 4, 5, 6, or 7 min in a 6-well plate. Immediately after NTP exposure, 100 μL of the viral inoculum (MOI 1) was added to confluent HaCaT cells in a 96-well plate. After a 1 h infection incubation, the inoculum was removed, and the cells were supplemented with 100 μL DMEM10 and incubated at 37 °C and 5% CO_2_ for 24 h.

The experiments involving NTP and infectious material were conducted as approved by the Drexel University Institutional Biosafety Committee in Biosafety Level 2 biosafety cabinet.

### 4.5. Quantification of HSV-1-Infected Cells

A recombinant strain of HSV-1 (KOS-GFP-HSV-1) was used for all experiments. This strain facilitated the detection of infected cells via GFP reporter expression (Figure 5A). At 24 h post-infection, the Cytation 5 was used to image and quantify GFP-expressing cells (Figure 5B). Image analysis was performed using the Gen5 program (version 3.11) (Agilent Technologies, Santa Clara, CA, USA). Images were acquired using a 4x magnification lens with the brightfield high contrast and GFP channels. Total cells, GFP+ cells, and the mean fluorescent intensity (MFI) of each well were quantified from images captured at 4x magnification using the Cytation 5. The percentage of GFP+ cells was calculated by dividing the number of GFP+ cells by the total cell count for each image.

### 4.6. Determination of HaCaT Cell Viability

Twenty-four hours post-exposure to NTP, cells were stained with 1 μg/μL propidium iodide (PI). The detection and quantification of PI-positive cells were accomplished using the Cytation 5. Image analysis was performed using the Gen5 program. Images were taken at 4x magnification, and the PI channel was used to measure fluorescence. The number of PI-positive cells and total number of cells in each well were used to calculate percent viability. Viability calculations were normalized to those of uninfected HaCaT cells not exposed to NTP.

### 4.7. Quantification of Hydrogen Peroxide and Nitrite

To determine the concentrations of hydrogen peroxide and nitrite after NTP application, phenol red-free DMEM10 (used to avoid interference with absorbance readings) was exposed to NTP in the absence of cells or in the presence of HaCaT cells plated at the same seeding densities used in the protocol for each device.

The NTP exposure procedures were device dependent. For the FE-DBD, 100 μL of phenol red-free DMEM was aliquoted into each well of a 12-well plate, exposed to NTP at 400, 500, and 1000 Hz, and supplemented with 900 μL of phenol red-free DMEM immediately after exposure. For the 2D-DBD, 100 μL of phenol red-free DMEM was added to every well in each 96-well plate. NTP exposure was conducted by centering the 2D-DBD electrode housing over a 3 × 3 grid of nine wells. Immediately after NTP exposure, the medium was transferred to the assay plate.

The Spectroquant hydrogen peroxide test procedure (Millipore Sigma; Burlington, MA, USA, Cat. 118789) and the Invitrogen Griess Reagent procedure (Invitrogen; Waltham, MA, USA, Cat. G7921) were used for the quantification of hydrogen peroxide and nitrite, respectively. All quantification assays were performed according to the manufacturers’ instructions. Absorbance was read using the Cytation 5, and the data were exported to an Excel file. Based on the respective line equation calculated from the standard curve, absorbance readings were used to determine the concentrations of hydrogen peroxide and nitrite in μM. To account for kit error, the concentration values of the NTP-exposed samples were subtracted by the concentration values of their respective no-NTP controls [61].

### 4.8. Statistical Analysis

A one-way ANOVA was performed for the analysis of the cell-free HSV-1 studies. All the other experiments were analyzed using a two-way ANOVA to determine statistical significance. Statistics were deemed significant if *p* < 0.05. All data were plotted and analyzed using GraphPad Prism version 9.5 (GraphPad Software, Boston, MA, USA).

## 5. Conclusions

We have provided the first results of investigations into an NTP-based therapy for herpes labialis and the management of cold sore outbreaks. Our in vitro results imply that the antiviral effects of NTP applied to a herpetic lesion will reduce productive viral replication, limit the spread of HSV-1 to nearby cells, and alter the susceptibility of uninfected cells. Furthermore, our results suggest that the antiviral effectiveness will be dependent on the NTP-generating device used for treatment and should be a factor in selecting the appropriate treatment modality for clinical use.

## Figures and Tables

**Figure 1 ijms-25-03839-f001:**
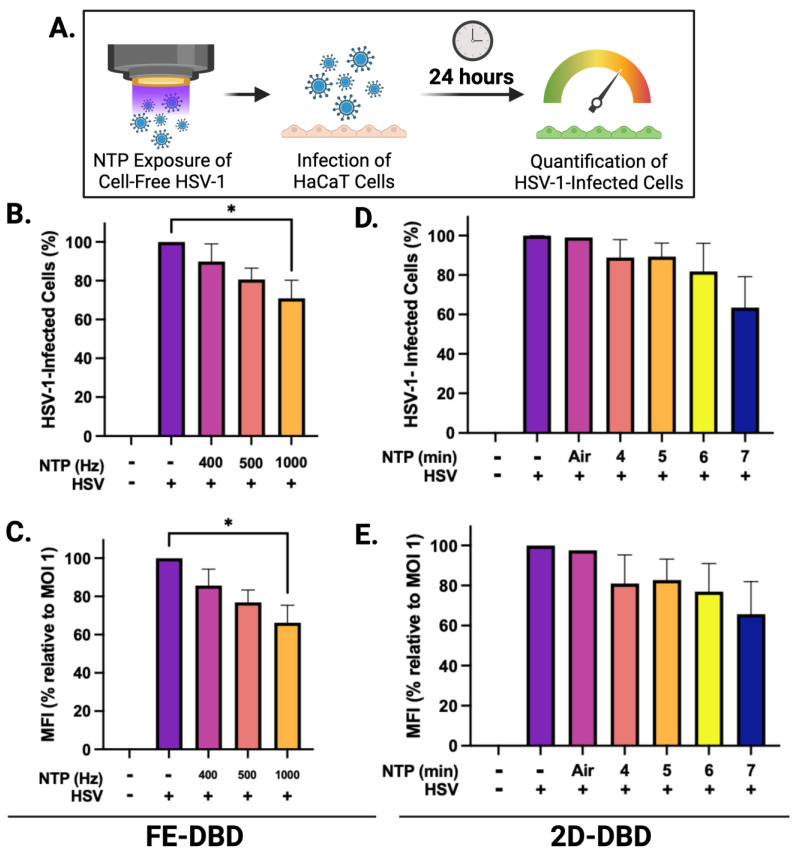
FE-DBD and 2D-DBD reduced HSV-1 infectivity and gene expression in cells. (**A**) Schematic of experimental timeline: Preparations of cell-free HSV-1 in media were exposed to NTP and then used to infect HaCaT cells immediately after NTP exposure. GFP expression was measured 24 h post-infection. (**B**,**C**) Frequency-dependent decreases in the percentage of infected cells and decreases in MFI after infection with FE-DBD-exposed cell-free HSV-1. (**D**,**E**) Exposure duration-dependent trends toward reduced infectivity and MFI were observed after 2D-DBD plasma application to cell-free HSV-1. * *p* < 0.05.

**Figure 2 ijms-25-03839-f002:**
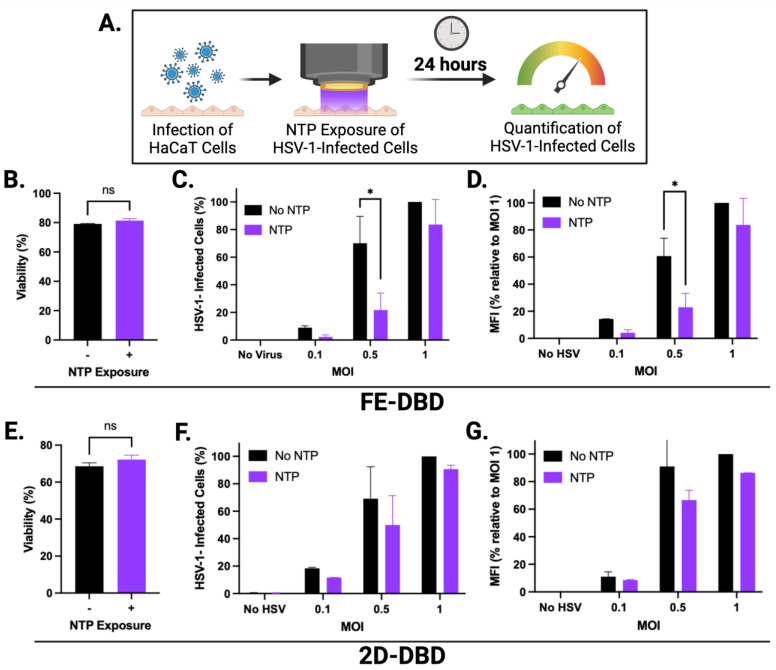
FE-DBD and 2D-DBD reduced number of HSV-1-infected cells 24 h post-NTP exposure. (**A**) Schematic of experimental timeline: HSV-1-infected HaCaT cells were exposed to NTP and measured for GFP expression 24 h later. (**B**) At 400 Hz, FE-DBD did not significantly impact viability of uninfected HaCaT cells. (**C**) Exposure of HSV-1-infected HaCaT cells to NTP at 400 Hz produced by FE-DBD resulted in reduced number of infected cells 24 h post-infection. Infected cells were quantified by GFP expression. (**D**) FE-DBD also decreased MFI of cells infected with HSV-1 24 h post-NTP exposure. (**E**) At 7 min NTP exposure using 2D-DBD, there were no significant changes in HaCaT cell viability. (**F**) 2D-DBD reduced number of infected cells and MFI within cells 24 h after their exposure to NTP (**G**), but this reduction was not statistically significant. Percentage of infected cells in (**C**,**F**) were calculated relative to number of HSV-1-infected cells detected after infection at 1 MOI without subsequent NTP application. * *p* < 0.05, ns = not significant.

**Figure 3 ijms-25-03839-f003:**
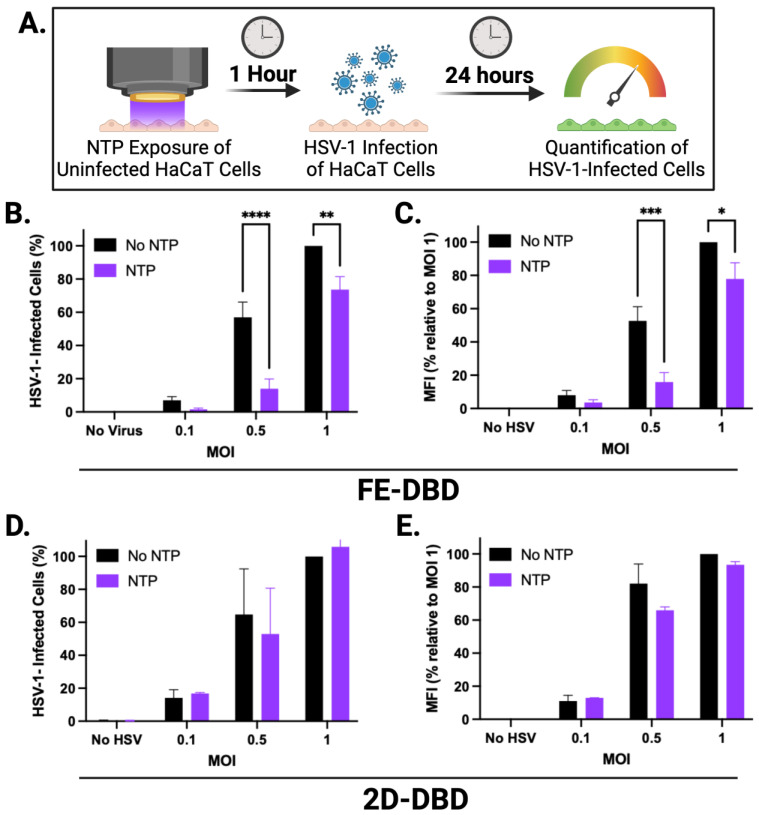
FE-DBD reduced susceptibility of uninfected cells to infection with HSV-1. (**A**) Schematic of experimental timeline: Uninfected HaCaT cells were exposed to NTP and then infected with HSV-1 1 h later. GFP expression was measured 24 h later. (**B**) NTP exposure of uninfected cells to FE-DBD plasma reduced infection by HSV-1 1 h post-exposure at three MOIs. (**C**) NTP exposure of uninfected cells also reduced levels of HSV-1 viral gene expression subsequent to infection. (**D**) 2D-DBD had no significant impact on susceptibility of cells to HSV-1 infection or (**E**) in MFI measured 24 h post-infection. Percentage of infected cells in (**B**,**D**) were calculated relative to number of HSV-1-infected cells detected after infection at 1 MOI without pre-exposure to NTP. **** *p* < 0.0001, *** *p* < 0.001, ** *p* < 0.01, * *p* < 0.05.

**Figure 4 ijms-25-03839-f004:**
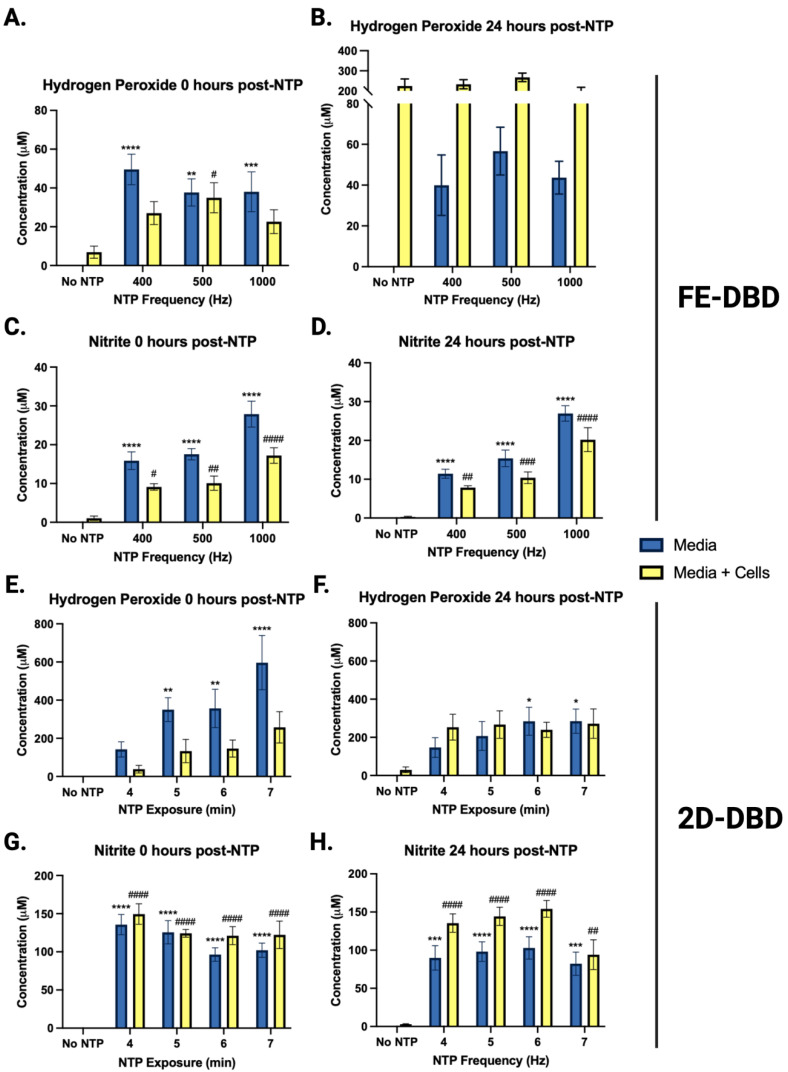
NTP increased hydrogen peroxide and nitrite concentrations in media. Hydrogen peroxide and nitrite concentrations were measured in media with and without HaCaT cells immediately after and 24 h following exposure to NTP generated by (**A**–**D**) FE-DBD or (**E**–**H**) 2D-DBD. Statistical analyses were applied to facilitate comparisons of concentrations measured in media only with or without NTP application (denoted with one or more *) or concentrations measured in media with cells with or without NTP application (denoted with one or more #). ****/#### *p* < 0.0001, ***/### *p* < 0.001, **/## *p* < 0.01, and */# *p* < 0.05.

**Figure 5 ijms-25-03839-f005:**
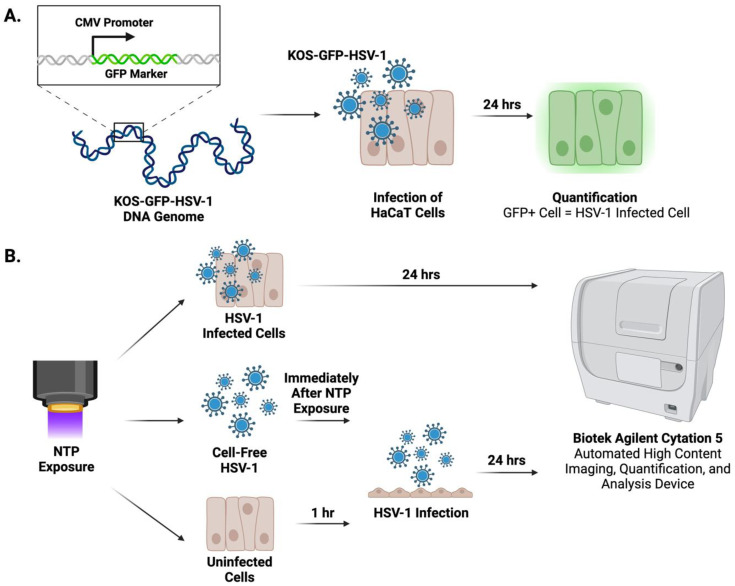
Experimental Design. (**A**) A KOS-GFP-HSV-1 strain was used to infect the HaCaT cells. HSV-1-infected cells were quantified 24 h post-infection by GFP expression. (**B**) Overview of the in vitro model of HSV-1 epithelial infection and NTP application. Cell-free HSV-1, HSV-1-infected cells, or uninfected cells were exposed to NTP. The antiviral effects were assessed in situ at 24 h post-infection on an automated Agilent Biotek Cytation 5 high-content imaging system.

**Figure 6 ijms-25-03839-f006:**
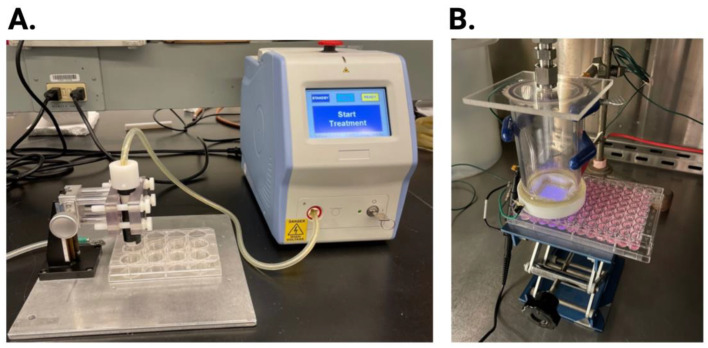
NTP devices. NTP was generated using an FE-DBD electrode (**A**) or a 2D-DBD electrode with a continuous air flow (**B**). In each device configuration, the distance from the electrode to the bottom of the plate was controlled mechanically.

## Data Availability

Relevant data are contained within the article.

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
