# Peer review of "Non-Thermal Plasma Reduces HSV-1 Infection of and Replication in HaCaT Keratinocytes In Vitro"

_ijms, 2024, doi:10.3390/ijms25073839_

Round 1

Reviewer 1 Report (New Reviewer)

Comments and Suggestions for Authors

This manuscript has evaluated the potential effect of non-thermal plasma (NTP) on HSV-1 infection in an HaCaT keratinocyte cell line. The authors’ data indicated that NTP not only interferes with the infectivity of HSV-1 and the viral replication in target cells, but also impacts the susceptibility of non-infected cells. It could be an interesting and promising methodology of NTP treatment for diseases especially viral infection, however, the current data depicted in this manuscript were not sufficient or defective to showcase the authors’ conclusion. More carefully-designed experiments and addition evidence should be provided in order to support the authors’ hypothesis.

1 In addition to HaCaT, the only cell line used in study, some key experiments should be repeated in other types of cells, such as neuronal cells. Considering the persistence of HSV-1 infection, the impact of NTP on viral lytic reactivation and reservoir size also needs to be tested.

2 The conclusion in lines 131-134 was not convincing. Without strong evidence, the direct influence of NTP on viral gene expression was still unclear, since the impaired binding and entry also attenuate the post-entry replication steps.

3 Does NTP show impact on cell-to-cell infection of HSV-1? In the section of 2.3 (page 6), NTP treatment significantly decrease the susceptibility of HSV-1 de novo infection in cells, but a cell-to-cell spreading assay should also be considered.

4 Have you compared the efficiency of RONS generation between FE-DBD and 2D-DBD?

5 In Figure 4B, the non-NTP treated media+cells group produced comparable level of hydrogen peroxide after 24h as NTP-treated groups, while in Figure 4F, media+cells didn’t show much production of hydrogen peroxide without NTP. Please explain such inconsistency.

6 If the antiviral activity of NTP depends on the induction of RONS, the low dose of RONS may not be functionally sufficient to inhibit HSV-1, while the excessive RONS might be detrimental by inducing the damage of nucleic acids, proteins, lipids, or physiological pathways. Above point needs to be further discussed in the manuscript.

Author Response

Reviewer 2 Report (New Reviewer)

Comments and Suggestions for Authors

In this manuscript, the authors investigated the effect of non-thermal plasma (NTP) on HSV-1 infection in vitro. They showed that NTP reduces the infectivity of cell-free HSV-1, reduces viral replication in HSV-1-infected cells, and diminishes the susceptibility to HSV-1 infection. These results are interesting and may provide a therapeutic approach for HSV-1-related epithermal diseases. There are several issues as follows:

1, the authors are suggested to detect viral titers to evaluate the inhibitor effect of NTP on viral infection, and perform RT-qPCR and western blot to evaluate viral gene/protein expression.

2, additional experiments are required to confirm that RNOS exerts the protective effect of NTP on HSV-1 infection.

Comments on the Quality of English Language

Not applicable

Round 2

Reviewer 1 Report (New Reviewer)

Comments and Suggestions for Authors

The revised manuscript has explained the major points and addressed some of my concerns. Even though some necessary experiments were missing to further support conclusions, the idea is conceptually interesting and promising, and this study is a good proof of principle for the potential application of NTP for treatment of viral infection. Overall, the revised manuscript can be considered for acceptance.

Reviewer 2 Report (New Reviewer)

Comments and Suggestions for Authors

The author's explanation partially answered my question. I have no further questions.

This manuscript is a resubmission of an earlier submission. The following is a list of the peer review reports and author responses from that submission.

Round 1

Reviewer 1 Report

Comments and Suggestions for Authors

In this study, Julia Sutter et al., have studied the investigations of NTP-based therapy for herpes labialis and the management of cold sore outbreaks. They provided experimental results that highlight non-thermal plasma (NTP) as a new alternative therapy for HSV-1 infection that would resolve cold sores faster and reduce the establishment of latent infection. They also investigated the antiviral effect of NTP using an in vitro model for HSV-1 cold sores, involving the application of NTP from two separate devices to cell-free HSV-1, HSV-1 infected cells, and uninfected cells. While the manuscript is clear in most parts, some minor revisions, if necessary can be incorporated into the manuscript.

1)    Could you clearly describe the differences between cell-free HSV-1 and uninfected cells.

2)    It is highly suggested to reduce some key words or use their abbreviated forms.

3)    There is a need to describe a brief overview of the morphology of HSV-1 viruses.

4)    You have used very few references in the discussion, especially between lines 429-483. This section needs more references to strengthen this study.

5)    While reading the discussion, it looks like I am a reading result of this manuscript. So, there is a need for improvement in this section.

6)    Prospects should be written logically and should be concise rather than adding so many sentences.

Author Response

Please see our response in the attached document.

Reviewer 2 Report

Comments and Suggestions for Authors

The manuscript “Non-thermal plasma reduces HSV-1..." by Sutter et al. describes the use of non-thermal plasma to inhibit the infectivity and replication of HSV-1 and the susceptibility of cells in culture to HSV-1. The antiviral activity of NTP is attributed to the generation of reactive oxygen and nitrogen species. The manuscript is generally well written.

The main problem with the study is its limited scope and relevance. The practical utility of such an approach is limited. The application of NTP to free viruses might have some application in the inactivation of viruses in fluids (blood), but the degree of inhibition is low.

The authors emphasise in the introduction and elsewhere the need to develop novel antiviral drugs and approaches to limit productive HSV-1 infection and also to limit the establishment of latency and potential reactivation (paragraphs 47-55); however, their approach is no different from other approaches that limit replication and then the establishment of latency reservoirs. Nevertheless, it will be used primarily in patients with recurrent disease who already have established reservoirs.

The use of NTP reduces viral infectivity by up to 40%, which is quite inefficient from a viral perspective (i.e. 10x6 to 6x10x5). There are numerous other drugs and approaches that are far more efficient. Furthermore, there are no replication assays to determine the exact effect of NTP on viral replication. Expression of GFP under a strong CMV promoter could be deceptive (non-replicative viruses could express GFP).

In the discussion section, the authors mention that NTP affects the expression of IE genes, which is not supported by the evidence currently presented. 

It would be useful to stress out the advantage of this approach compared to others (i.e. photodynamics).

Minor:

-       -  the in vitro model does not represent the model of cold sores, but a model for the replication of HSV-1

-        - Figure schematics under A) in all figures are confusing and not easy to understand which picture represents what. Time of NTP application is important and clock with 3 am is distractive.

Author Response

(The authors gave the same response as above.)
